# Increased fire activity under high atmospheric oxygen concentrations is compatible with the presence of forests

Rayanne Vitali [1] ✉, Claire M. Belcher[1], Jed O. Kaplan [2] & Andrew J. Watson [1]

Throughout Earth's history, the abundance of oxygen in our atmosphere has varied, but by how much remains debated. Previously, an upper limit for atmospheric oxygen has been bounded by assumptions made regarding the fire window: atmospheric oxygen concentrations higher than 30–40% would threaten the regeneration of forests in the present world. Here we have tested these assumptions by adapting a Dynamic Global Vegetation Model to run over high atmospheric oxygen concentrations. Our results show that whilst global tree cover is significantly reduced under high $O_2$ concentrations, forests persist in the wettest parts of the low and high latitudes and fire is more dependent on fuel moisture than $O_2$ levels. This implies that the effect of fire on suppressing global vegetation under high $O_2$ may be lower than previously assumed and questions our understanding of the mechanisms involved in regulating the abundance of oxygen in our atmosphere, with moisture as a potentially important factor.

Life on Earth has been defined by the concentration of oxygen in the atmosphere. It allows us to breathe today, was a critical influence in the evolution of plants and animals and plays an important role in the Earth system and its geochemical cycles[1–3]. For much of Earth's history the atmosphere had only trace levels of oxygen, until around 2.4–2.5 billion years ago when substantial free oxygen appeared, in a shift termed the great oxidation[4]. However, whilst estimates on the lower limit of atmospheric oxygen since the appearance of land plants approximately 420 million years ago (Ma) are fairly robust[5–10], estimates of the upper limit of atmospheric oxygen and processes involved in its regulation are still poorly understood[11].

The presence of fossil charcoal in sedimentary rocks since the late Silurian has been used not only to indicate the occurrence of wildfires throughout subsequent evolutionary history, but also to put constraints on the variability of atmospheric oxygen concentrations[12,13]. For a sustained fire to occur, three basic elements are required: a source of ignition such as lightning, fuel that can burn, and a supply of oxygen[14]. It is likely that natural ignition sources such as lightning strikes have always been present on Earth[5,15] and adequate fuel has been available since the evolution of land plants around 420 Ma[12]. Thus

evidence of fire occurring (e.g., fossil charcoal) from this time onwards indicates that atmospheric oxygen concentrations must have been within the bounds that support natural fire, termed the fire window by some[6,16]. Subsequently, variations in the concentration of oxygen in the atmosphere are accompanied by fluctuations of the flammability of our planet throughout time[6,17,18].

The lower limit of the fire window has been investigated in numerous experimental studies[5–10]. Combustion experiments by Watson[8] found that dry paper was unable to ignite below 17% vol. $O_2$ and experiments using more realistic fuels such as wood, peat, and other plant materials show similar results[9]. Similarly, Belcher et al.[6] found dry peat could not ignite at 15% vol. $O_2$ and that fire was greatly supressed below 18.5% vol. $O_2$. Hence, the occurrence of charcoal in the fossil record from 420–405 Ma and its near continual presence in sedimentary rocks since the emergence of land plants around 370 Ma indicates that atmospheric oxygen must have surpassed this lower limit since this time[5,10].

The upper limit of atmospheric oxygen experienced throughout Earth's history is harder to determine and there is no completely model-independent estimate of what this might be, with most

[1]Global Systems Institute, University of Exeter, Exeter, UK. [2]Department of Earth Sciences and Institute for Climate and Carbon Neutrality, The University of Hong Kong, Hong Kong SAR, China. ✉e-mail: rv237@exeter.ac.uk

geological evidence of atmospheric oxygen only indicating its presence or absence[5,11]. The closest we have come to direct evidence of high oxygen concentration is in the form of fossil air trapped in amber samples from the Cretaceous period, which measured 30% vol. $O_2$[19]. However, the samples have since been widely questioned[20], for instance concerns as to how effectively amber bubbles can create an air-tight seal, potentially causing distortion in the composition of the samples over long time periods, leading to a near-universal rejection of such measurement[20,21]. Other biological evidence includes the fossilised remains of giant winged insects and other fauna, insects, spiders, millipedes, and primitive amphibians dating to a 50-million-year period in the late Carboniferous[22–25]. It has been proposed that the existence of giant dragonfly-like insects (Meganeura) require atmospheric oxygen concentrations to be greater than present in order for them to survive at such large sizes[26,27]. Again, however, there have been questions surrounding insect oxygen toxicity and trachea efficiency which dispute these claims[28].

A compelling argument for the upper limit of atmospheric oxygen comes from the fact that, once established in the Phanerozoic, forests persisted throughout periods of purportedly high $O_2$ concentrations[29]. Several combustion experiments have investigated the relationship between oxygen concentration and flammability of vegetation-derived fuels[8,10,30]. Such studies indicate that the probability of ignition of a fire increases sharply with increasing oxygen concentrations[6,8,10,30]. Whether an ignition can lead to sustained combustion (e.g., spread) is determined by various factors including temperature, fuel moisture, wind speed, fuel density etc.[31,32]. Of these, moisture content is thought to play a critical role[10,32]. Whilst dry fuels enable fire spread, if fuel moisture content surpasses a certain threshold, fire cannot be sustained; this is defined as the moisture of extinction. The value of moisture of extinction for different natural fuels has been found to increase with oxygen concentration in combustion experiments[10,30], meaning wetter fuels can carry fire at oxygen levels higher than the present 21%. Furthermore, the concentration of atmospheric oxygen can also influence the energy released from a fire[33,34]. Heat of combustion (HoC) is defined to be the total amount of energy released in a fire in the form of heat[35] and has been found to vary across not only different vegetation types[33,35,36] but also across different oxygen concentrations[33,34]. A rise in atmospheric oxygen is therefore likely to be accompanied by increasing wildfire activity. The fossil evidence for the continued presence of forests since ~370 Ma[18,37], suggests that atmospheric oxygen concentrations could never have risen so high that the frequency and spread of fires greatly suppressed vegetation and prevented the regeneration of forests[29,38]. However, what level of oxygen this would occur at is debated.

Early calculations from Watson et al.[29] concluded that with 25% vol. $O_2$ forest regeneration would be prevented by continuous fire, even at elevated moisture levels. Others have suggested that forests with higher moisture content and rapidly reproducing trees would be more tolerant to rising oxygen concentrations, arguing for a higher upper limit of 30% vol. $O_2$[5,12,38]. Whilst it is generally agreed that fires are more likely to ignite under high oxygen concentrations, it has been argued that rate of fire spread is strongly dependent on fuel moisture[10,39]. Experiments by Wildman et al.[10] found that vegetation-based fuels with high fuel moisture were unable to burn, even in high oxygen concentrations of 35% vol. $O_2$, suggesting that atmospheric oxygen concentrations of this level and above could be compatible with the existence of forests[10,39]. Therefore, the upper limit of the fire window is still largely unknown and could be between 25–35% vol. or potentially higher. Yet, to date, no study has thoroughly examined the hypothesis of the upper limit of the fire window: that high atmospheric oxygen concentrations could have led to widespread wildfires that may have inhibited the growth of forest and potentially the formation of forest biomes, and whether fuel moisture might mitigate such effects.

Geochemical models have been used to try and gain an understanding regarding the bounds of atmospheric oxygen through time and have also led to mixed results[40–44] but are largely dependent on whether or not fire feedbacks are included. Regulation of atmospheric oxygen within a proposed fire window suggest that mechanisms have been in place over 100 s of millions of years that prevent concentrations decreasing past the lower limit or increasing further than the upper limit. Such mechanisms must achieve regulation through a series of negative feedback loops which likely involve sources or sinks of oxygen to the atmosphere, which over the long-timescales of interest are predominantly linked to organic carbon burial (long-term oxygen source) and oxidative weathering (oxygen sink)[5,38,45]. At present-day levels of atmospheric oxygen, it is believed that oxidative weathering goes nearly to completion[5,46] and so has diminishing power under high atmospheric oxygen concentrations and therefore cannot explain regulation to counteract rising levels[5,47]. Instead, processes involving mechanisms that change organic carbon burial would be needed to provide regulation. Ocean-based feedbacks have been proposed to keep atmospheric oxygen levels above the lower-limits of the fire window[5,48] but this fails at regulating high oxygen levels[5,45,49]. Fire itself via its response to oxygen[10,30,50] has been suggested to impact the amount of organic carbon burial through suppression of vegetation and its subsequent influence on the abundance of phosphorus, a key limiting nutrient[5]. It has been suggested by Kump[45] that oxygen enhancement of fires would result in more phosphorus being redirected from the land to the ocean, reducing the overall carbon-phosphorus burial ratio in ocean sediments (where there is a lower C:P ratio in marine organic matter than terrestrial). This would reduce carbon burial which is the long-term source of atmospheric oxygen[45]. Another fire feedback was proposed by Lenton and Watson[38] in which increases in atmospheric oxygen limit the biomass of land vegetation through increased fire activity, shifting ecosystems from forests to faster-regenerating vegetation with lower biomass[5,18]. This loss of deep rooting ecosystems suppresses phosphorus weathering by land plants, and reduced primary productivity of vegetation lowers both land and ocean-based carbon burial[5,38].

Such negative fire feedback have been incorporated into biogeochemical models that predict the abundance of oxygen in Earth's atmosphere over hundreds of millions of years. However, the exact mechanisms and what the overall strength of feedback is still widely disputed[5,42,51,52] and lead to large differences in estimates of the fluctuations of atmospheric oxygen throughout the Phanerozoic (the past 550 Ma)[40–44,51,52]. These estimates are produced using relatively simple box models, some of which have considered how wildfire feedbacks to plant growth are linked to the regulation of atmospheric oxygen[42–44,49]. Berner and Canfield[51] were among the first to use a mathematical model to simulate the evolution of atmospheric oxygen through time. They suggested a large peak in atmospheric oxygen around 300 Ma, reaching 35% vol. $O_2$ which remarkably coincides with the fossilized remains of giant insects[22,23] and increased abundance of charcoal, a product of wildfires, in the sedimentary record[17]. This signature oxygen peak has since been broadly replicated in subsequent biogeochemical models such as GEOCARBSULF[40] and the Carbon-Oxygen-Phosphorus-Sulphur-Evolution (COPSE) model[42,43]. However, such models include various assumptions and uncertainties which allow room for disagreement, especially regarding the upper limit of the oxygen peak. For example, an error analysis presented by Royer et al.[53] found that through using a larger range of input data in a more recent version of GEOCARBSULF[54], the atmospheric oxygen peak could reach levels greater than 40% vol. $O_2$. Although this study used an upper limit of 50% vol. $O_2$ based on constraints of plant flammability studies and geological record of wildfire, the model does not include representation of fire effects on vegetation or associated feedbacks[53]. All the models explore the relationship between sedimentary reservoirs of carbon, oxygen, and oceanic and atmospheric reservoirs. However, the

COPSE model[42] also includes the interaction of fire and its associated feedbacks to a basic representation of terrestrial primary productivity. In this case, fire and its feedback to forest extent becomes particularly important in preventing atmospheric oxygen from rising to levels greater than some of the suggested upper limits of the fire window[5,29]. Despite these long-standing (since 1978) assumptions regarding the upper limit of the fire window, no studies have assessed to what extent oxygen-driven wildfires suppress forest growth at high levels of oxygen nor how this might influence global tree cover. To achieve this, global projections of feedback by fire on forests are required not only to move debates and our understanding forward but also to build the next generation of biogeochemical models that understanding the regulation of oxygen over Earth's long history.

The large-scale interactions of wildfire on land plants has been the subject of numerous studies[13,55,56] using Dynamic Global Vegetation Models (DGVMs). Such models have been used to address questions such as what earth's vegetation would look like in a world without fire[57] and the influence of fire on the distribution of biomes in the present-day[31,58], the future (based on climate projections)[59], and the historical past[60].

In this work, to investigate the effects that fire has on global vegetation under the upper limits of the fire window, we run simulations using the LPJ-LMfire DGVM[61,62]. In order to run simulations for varying atmospheric oxygen levels, novel enhancements are made to the fire module within LPJ-LMfire to account for the effect of oxygen concentration on fire. To achieve this, we expand three core parameters to vary over atmospheric oxygen: probability of ignition, moisture of extinction, and heat of combustion, that determine the likelihood of ignition, the number of fires, the resultant fire behaviour (rate of spread and fireline intensity), and the burned area which together, influence the abundance of tree cover in the model. We then use LPJ-LMfire to test the hypothesis that the regeneration of forests would be at risk under $O_2$ concentrations greater than 30% vol.[29]. We show that whilst our simulations indicate enhanced fire frequency and burned area, forest cover persists globally even in simulations with 35% vol. $O_2$. This result implies that not only could the upper limit of the fire window be higher than previously considered, but also fire feedback to atmospheric oxygen may be weaker. We find that the fire suppression on present-day forest cover is approximately 26% when compared to a world without fire case (<17% vol. $O_2$), almost half of the 50% reduction estimated by Bond et al.[57] who found that present-day forest cover would double in a world without fire. Yet still significantly greater than initial estimates of 5% reduction of vegetation biomass under present atmospheric levels (PAL) compared to the no-fire scenario used in previous versions of biogeochemical models[5,42], supporting that the effects of fire feedback on atmospheric oxygen concentration may be substantial.

## Results & discussion
### Effects of atmospheric oxygen on fire and global vegetation
Watson et al.[29] suggested that even a rise to 25% vol. $O_2$ would mean that lightning could cause a fire, even if accompanied by rain, with Lovelock[2] further stating that this would result in little or no forests as neither tropical forest, or arctic tundra would be able to cope with resulting widespread fires. A slightly higher estimate is given by Lenton[5] who argues that this value of 25% ought to increase to around 30% vol. $O_2$ when considering the sensitivity that fire has to fuel moisture. These assumptions are broadly based on the combustion experiments undertaken by Watson[8], but without incorporating the results into a vegetation model. Equations resulting from this work for the probability of ignition and the moisture of extinction are described by Watson and Lovelock[30] and these are what we used to adapt the fire module of LPJ-LMfire to account for changing atmospheric oxygen concentrations (see methods). By incorporating these long-standing experimentally derived relationships into a DGVM with a coupled fire module we have been able to provide for the first time a quantitative estimate of global tree cover as influenced by oxygen-driven changes in fire. As anticipated, simulations run at 35% vol. $O_2$ show a significant reduction in global tree cover when compared to present-day levels of atmospheric oxygen (Fig. 1). This is due to an increase in the probability of ignition and a change in fire behaviour, which increase fire frequency and burned area. However, unlike some of the previous qualitative estimates[5,8,29,63], we find that whilst tree cover is greatly reduced, areas of forest (tree cover > 60%) still exist globally, suggesting that such an increase of atmospheric oxygen remains insufficient to prevent the regeneration of forests due to very high fire frequencies and large burned areas. This challenges the previous qualitative estimates of the upper limit of the fire window.

Simulations ran with fire switched off show the vegetation potential in a world without fire and can therefore be used to compare differences in the suppression of vegetation and forest cover due to fire under different atmospheric oxygen concentrations. We found that the reduction in global forest cover due to fire suppression (percentage decrease from forest potential with no fire) increases from 26% to approximately 60% for levels of 20.95 and 35% vol. $O_2$ respectively (see Fig. 2b). For present-day levels of atmospheric oxygen, several studies have examined the role of fire in influencing patterns in global biogeography. Bond et al.[57] ran simulations for a world without fire using the Sheffield DGVM, excluding anthropogenic activity, and found forests cover to be reduced by around 50% at present due to fire. This is significantly greater than our simulations, which show around a 26% reduction in forest cover for present-day atmospheric levels. However, since Bond et al.[57] others have repeated the world without fire simulations using modern DGVMs with the average showing a much smaller 20% suppression on forest cover[64], agreeing more closely with our simulations. This suggests that the reduction in forest cover estimated by Bond et al.[57] was too high, and was potentially caused by the low spatial resolution of the model simulations. Consequently, the continuous presence of forests on Earth since the Devonian may not necessarily indicate that atmospheric oxygen concentrations were below 35%.

### The role of fuel moisture content
Results for atmospheric oxygen concentrations between 20.95 and 35% vol. $O_2$ show that whilst the total global number of fires increases sharply with oxygen concentration (Fig. 2a), the total burned area increase is much smaller, slower and starts to stabilize around 30% vol. $O_2$ at approximately 2.7 times global burned area at present-day concentrations (Fig. 2a).

Whilst fires are more likely to be ignited under high oxygen concentrations, fire extent is controlled by the rate of fire spread, which is argued to be more dependent on fuel moisture[10]. Combustion experiments by Wildman et al.[10] found that pine-needle litter beds of moisture contents >42%, which is realistic under rainy conditions or high humidities, were unable to sustain a spreading fire even at 35% vol. $O_2$. Furthermore, these experiments showed no change in fire rate of spread between oxygen contents of 21–35% vol. $O_2$. Our results show that whilst global rate of fire spread increases globally with increasing atmospheric oxygen concentration (Fig. 3 and black line Fig. 4b) results vary based on latitude and fuel moisture within the model. Whilst mid-latitudes show increased rate of spread, high and low latitudes with forest cover see little or no change between PAL (20.95% vol. $O_2$) and 35% vol. $O_2$, with rate of spread staying low (Fig. 3). This is due to high fuel bulk densities and perennially high fuel moistures attenuating fire rate of spread within the model. When considering different bands of fuel moisture, we found gridcells with the highest fuel moistures have smaller increases in the rate of spread with increasing oxygen concentration, with very little change observed at fuel moistures >60% (Fig. 4b). Thus, higher moisture contents of vegetation have the potential to counteract the effect of higher

a

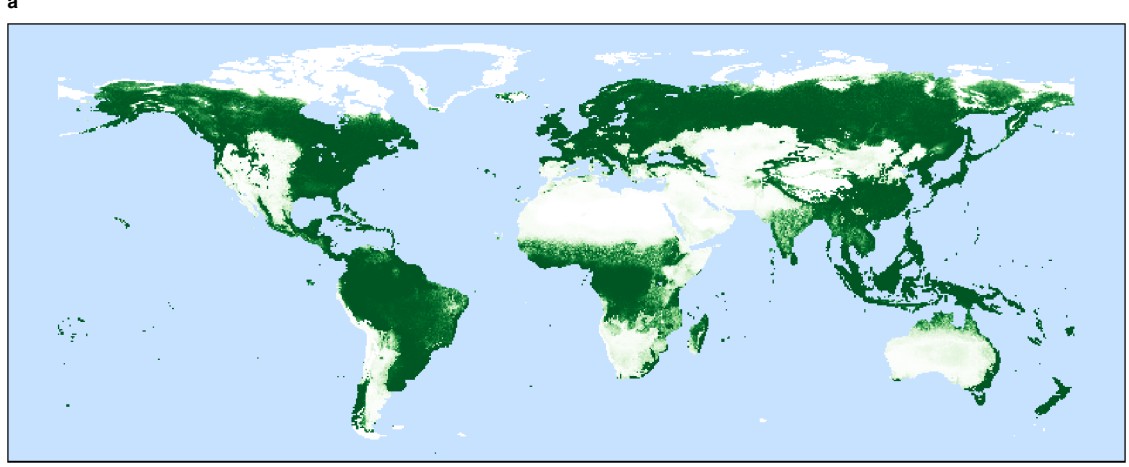

b

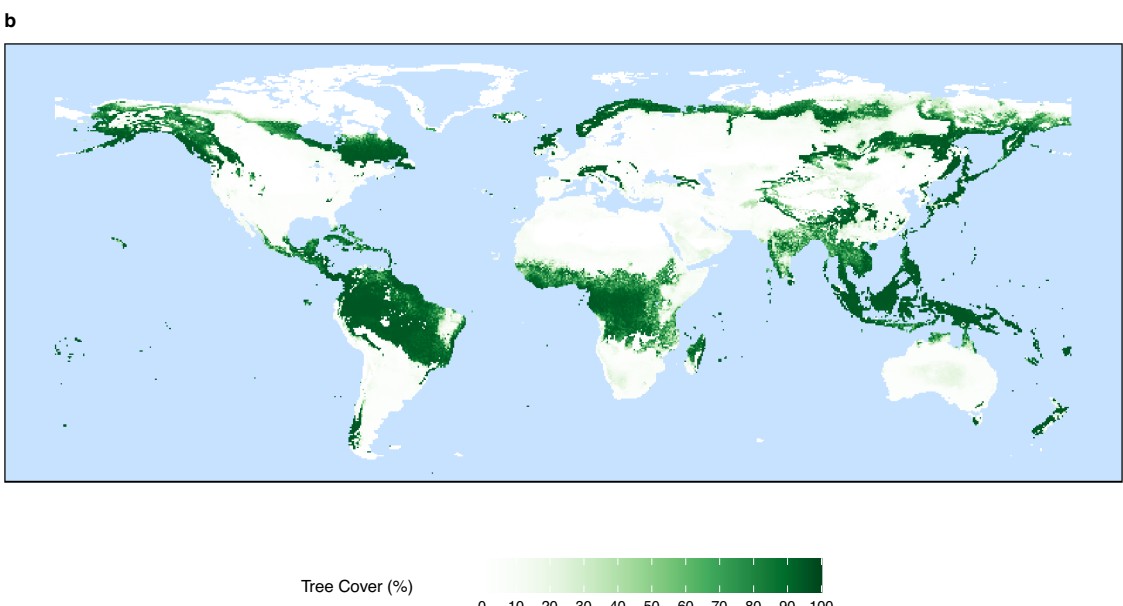

Tree Cover (%)

0  10  20  30  40  50  60  70  80  90  100

**Fig. 1 | Global tree cover.** Taken as a 10-year annual average from the last decade of LPJ-LMfire for **a** present-day level of atmospheric oxygen (20.95%) and **b** 35% vol. O₂.

atmospheric oxygen concentrations on the rate of fire propagation and may explain the diminished response of burned area in higher atmospheric oxygen concentrations, agreeing with the experiments conducted by Wildman et al.[10] and supporting those who have argued for higher limits (>35% vol. O₂) of atmospheric oxygen historically[10,39,53].

In our simulations with high atmospheric oxygen concentrations, forests still exist in areas with perennially high precipitation (Fig. 4a), where rate of spread becomes limited through fuel moisture as above, mainly at low and high latitudes. Low-latitude rainforests remain wet and humid year-round whilst low evaporative demand retains fuel moisture in high-latitude forests. This climate-limitation can be also seen in vegetated mountain ranges such as the Himalayas due to low temperatures at high altitudes (Fig. 4a). Therefore, in these regions, although under high oxygen concentrations the chances of a fire igniting are high since probability of ignition and moisture of extinction increases, fuel moisture content remains high enough to limit the rate of spread and prevent fires from becoming widespread (Fig. 4a). Hence why the number of fires igniting increases sharply with increases of atmospheric oxygen whilst burnt area appears to level off (Fig. 2a). Conversely, our results suggest that mid-latitude forests are more sensitive to rising atmospheric oxygen than the tropics (Figs. 1

and 3). This is due to the ignition efficiency in these regions increasing more with oxygen concentration than elsewhere because of increases in the probability of ignition, seasonally lower fuel moisture contents that allow high rate of fire spread as well as the inherent flammability of the regional vegetation. This difference reflects what is observed globally, where tropical regions such as Brazil have higher fuel moisture content than temperate forests for much of the year, particularly in closed-canopy situations[65].

From our results, we would expect that in a warmer, wetter world forests would be less flammable globally than a cooler drier world as the flammability of forests would be limited by fuel moisture content. This is result is supported by the observed reduction in Eocene charcoal compared to the preceding Palaeocene[17,18], which has been attributed to hydrological changes leading to increased rainfall[66] and the widespread establishment of rainforest biomes that were fire-resistant[18]. Additionally, biomass combustion records and charcoal morphologies indicate that a shift to increasingly colder and drier conditions from the late Miocene to Pliocene promoted the establishment of pyrophilic grasslands, where wildfire activity sustained flammable ecosystems[67]. Whether global forests could be totally eradicated by fire under high atmospheric oxygen concentrations also depends on factors such as resprouting and vegetative reproduction

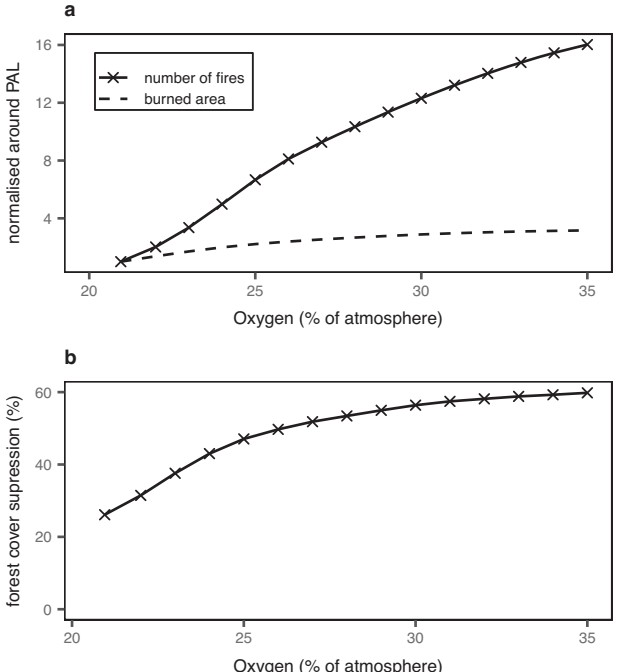

**Fig. 2 | Flammability and forest cover fire suppression output from simulations over atmospheric oxygen.** Output from LPJ-LMfire shown as total global 10-year averages over varying atmospheric oxygen levels showing **a** number of fires (solid line) & burned area (dashed line) and **b** forest cover suppression defined as the percentage decrease from a world with no fire.

as well as seasonality. A study conducted by Hollaar et al.[68] measuring smectite and kaolinite abundance from the Jurassic as proxies for seasonality and the hydrological cycle a strong correlation between lower humidity and greater fire activity. While these studies support our results, they leave open the question of whether moisture could sufficiently protect forest vegetation from fire even at very high concentrations of oxygen.

### Upper limits of the fire window & atmospheric oxygen

Though the lower bound of the fire window (the limits of atmospheric oxygen levels compatible with fire) is fairly robust based on evidence from combustion experiments[5–10], previous attempts to define the upper bound has relied on assumptions and simple calculations regarding the atmospheric oxygen concentration at which fires prevent the regeneration of global forests[5,11,29,30,38]. Our results show that increases in fire, even at high levels of atmospheric oxygen concentrations only reduces forest cover to approximately half of what it would be with no fire and so the upper limit of the fire window is at least greater than the previously assumed value of 35% vol. $O_2$. Furthermore, the suppression of forest cover by fire appears to stabilise (begins to plateau around ~60%) under high oxygen concentrations (Fig. 2b) and so at levels greater than 35% vol. $O_2$ we would expect the impact of fire to have limited further effects on forest cover. It seems that fuel moisture becomes the limiting factor. We argue, therefore, that the upper limit of the fire window is likely much higher than 35% vol. $O_2$, or possibly even unbounded.

Whilst the upper limit of the fire window is higher than previously thought, the upper limit of atmospheric oxygen itself and whether forests can exist under high oxygen concentrations over the Earth's history, relies on more complex mechanisms and the feedback at play. For instance, decreases in forest biomass under high atmospheric oxygen concentrations directly limits the amount of organic carbon available for carbon burial, the long-term source of $O_2$[5,38,45]. Several negative fire feedbacks have been

proposed that also lower rates of carbon burial under high oxygen levels such as reduced phosphorus weathering by roots as a result of fire suppression on vegetation, which in turn lowers productivity and hence carbon burial[5]. It is therefore possible that the 60% reduction in forest cover under 35% vol. $O_2$ (compared to no fire forest cover) is enough to slow carbon burial to an extent that the rise of atmospheric oxygen is too slow compared to the measures that counteract it, limiting the ability of the biosphere to sustain a high oxygen atmosphere. Whether or not atmospheric oxygen can surpass levels of 30/35% vol. $O_2$ thus relies on the strength of such geochemical feedback on oxygen concentrations. Other non-fire linked processes influence the concentration of atmospheric oxygen through time and may therefore have a more important role than previously considered. High atmospheric oxygen can limit productivity of plants through inhibiting $CO_2$ fixation through the Rubisco enzyme[69,70] which would further impact vegetation biomass and hence carbon burial. Other processes have the potential to counteract reductions in organic carbon burial under high atmospheric levels, such as continental uplift which can enhance carbon burial through increasing the flux of reactive phosphorus[38]. Periods of increased uplift could therefore lead to high atmospheric oxygen levels greater than 30% vol. $O_2$ being reached, as may have been the case during the Permian-Carboniferous[71].

### Implications & future directions

In order to gain insight into the history of atmospheric oxygen through time, we have to turn to biogeochemical models that run over large timescales such as COPSE[42,43] and GEOCARBSULF[40,41,71]. In these models, the fire-vegetation feedback is based on the very high sensitivity of forest cover to fire suggested by Bond et al.[57] and are evaluated against their ability to make atmospheric oxygen fall below the previously proposed fire window upper limit of 30% vol. $O_2$, which we have shown was likely not the case. Hence, it is plausible that atmospheric oxygen concentrations have been less controlled by fire-vegetation feedback and so could have reached levels even higher than previously simulated during the Permian-Carboniferous and the Cretaceous periods. Furthermore, the representation of the magnitude of fire activity in these models is taken as a simple function of oxygen concentrations and does account for changing vegetation, heat of combustion, or moisture content as we have shown here. A recent study by Belcher et al.[49] indicated that by including an evolving fire-vegetation feedback scenario in the COPSE model, driven by the rise of angiosperms, oxygen was better regulated, allowing for the emergence of closed-canopy angiosperm tropical rainforests —highlighting the importance of improving models to allow for changing relationship between vegetation, fire, and oxygen concentrations.

Our model simulations suggest that the assumptions made on the upper limit of the fire window do not hold because forests are able to persist in the present world under high atmospheric oxygen levels (>30% vol. $O_2$) and fuel moisture is likely to have been the controlling factor on fire frequency and intensity under high $O_2$ concentrations, which precluded the total eradication of woody vegetation by fire. This suggests we require a deeper understanding of the controls of atmospheric oxygen through time. Whilst it seems that fire may play a lesser role in setting an upper limit of atmospheric oxygen than previously thought, other processes of importance remain largely unknown.

Further experimental data is needed to look at the relationships between fire, natural fuels, fuel moisture and atmospheric oxygen, particularly at levels greater than 35% vol. $O_2$. This would give better insight into fire behaviour under high levels of atmospheric oxygen and would lead to greater understanding into the interactions between

a

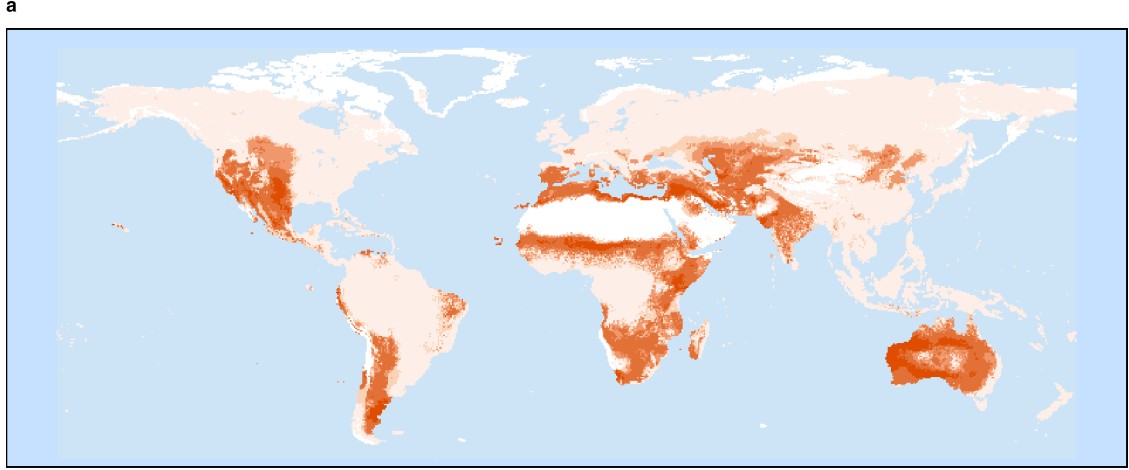

b

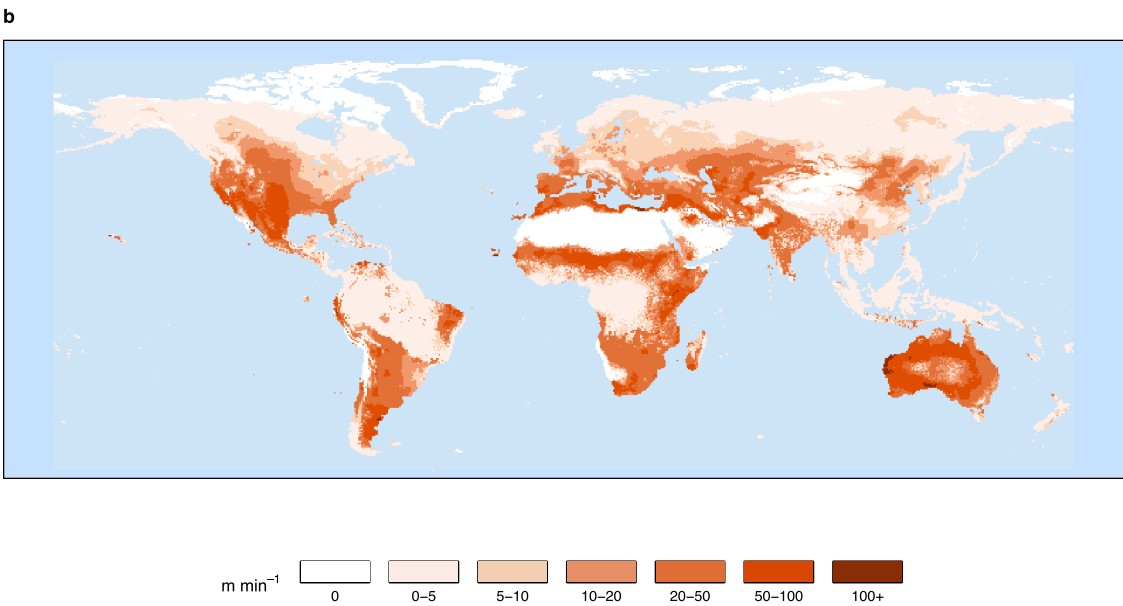

**Fig. 3 | Global fire rate of spread.** Taken as a 10-year annual average from the last decade of LPJ-LMfire for **a** present-day level of atmospheric oxygen (20.95%) and **b** 35% vol. O$_2$.

global fire and atmospheric oxygen levels through time. Results presented here have widespread implications on areas that rely on assumptions regarding levels of atmospheric oxygen concentration through time such as the evolution of land plants which are known to be tied to fire and atmospheric oxygen[72], understanding of the Earth system's ability to deal with ocean anoxia[49] and output of models which are used to simulate biogeochemical cycles over deep time[40–43,71]. From this, there is a clear need to gain a better understanding of the evolution of atmospheric oxygen through further developing biogeochemical models, particularly constraints on atmospheric oxygen and fire-feedbacks in accordance with our results. For example, using a biogeochemical model to test whether the biosphere can sustain a high oxygen atmosphere with a large reduction of forest cover and hence carbon burial. Whilst our research indicates that potential atmospheric oxygen concentrations may extend above that previously predicted for modern-day vegetation distributions and climate scenarios, our approach serves as blueprint for the need to develop a palaeo-DGVM that can run over deep time. This would need to include ancient plant functional types and paleoclimate scenarios if we are to fully understand fire-oxygen feedback to the Earth system over the long-term. It is clear, however, that the question of what the

upper limit of atmospheric oxygen level could be on a life-supporting Earth remains.

## Methods
### Model description
Here we used LPJ-LMfire, a revised version of the Lund-Potsdam-Jena (LPJ) DGVM[62] coupled to a modified fire module. The fire module is based on SPITFIRE (SPread and InTensity of FIRE) fire model[61] but with significant improvements in simulated burned area for natural fire due to the inclusion of the explicit calculation of natural ignitions, the representation of multi-day burning and coalescence of fires, and the calculation of rates of spread in different vegetation types[73]. We ran the model at a 0.5° spatial resolution, vegetation is represented in the model through 9 Plant Functional Types (PFTs) including 2 tropical trees, 3 temperate trees, 2 boreal trees and 2 grasses (see Supplementary Table 2) and the model operates on a daily timestep. For a full model description, see Pfeiffer et al.[73] LPJ-LMfire was built for simulating global fire-vegetation interactions during prehistoric and preindustrial times[62]. This model was therefore chosen to enable us to model interactions between oxygen, fire, and

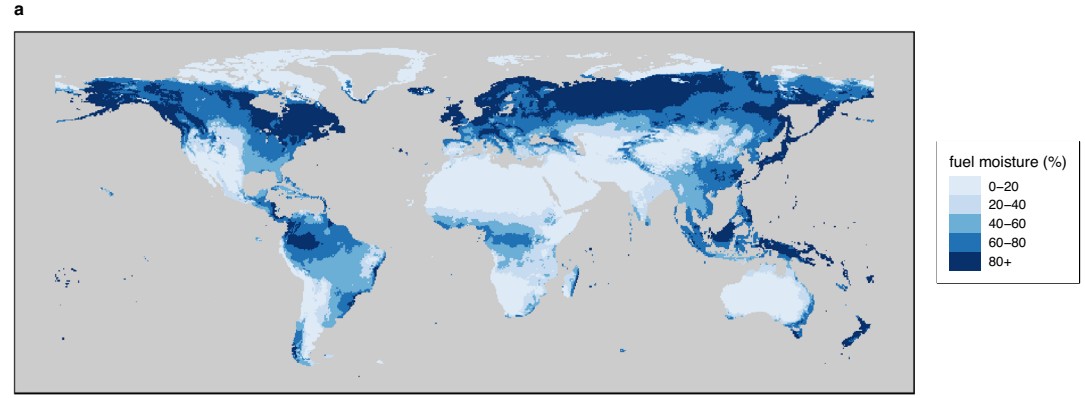

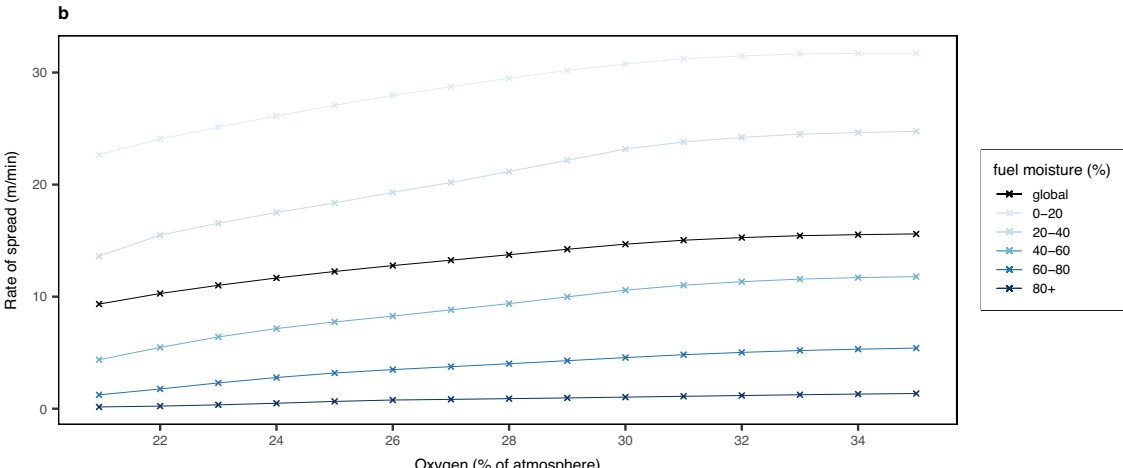

**Fig. 4 | Global fuel moisture bands.** Plots displaying global total fuel moisture content taken as a 10-year annual average from the last decade of LPJ-LMfire under 35% vol. $O_2$ (**a**) and total global 10-year averages of rate of fire spread over varying atmospheric oxygen levels for different bands of fuel moisture content (**b**).

vegetation without human influence (ignitions, land-use, climate change etc.), enabling us to investigate if increasing atmospheric oxygen concentration to the proposed upper limits of the fire window prevents forest regeneration in a present-day world as suggested[5,8,10,30,38,63]. An oxygen variable was added to the model in order to simulate varying atmospheric oxygen levels. We also altered the fire module in order to simulate the effects of changing atmospheric oxygen concentrations on fire behaviour in three main ways: probability of ignition, moisture of extinction and heat of combustion.

## Probability of ignition

Probability of ignition is defined to be a rating of the likelihood that a lightning stroke will ignite and produce a successful fire in dead, fine fuels[30,74]. Combustion experiments have shown that probability of ignition can increase greatly with rising atmospheric oxygen concentration[6,8,30]. Here we include an equation of probability of ignition as a function of atmospheric oxygen concentration ($O_x$) and fuel moisture content ($M$), taken from Watson and Lovelock:[30]

$$PI(O_x, M) = [308.02 - 27.406(O_x) + 0.634(O_x)^2 - 0.0044(O_x)^3] \ln(M) - 633.54$$
$$+ 42.327(O_x) - 0.2194(O_x)^2 - 0.0075(O_x)^3$$
$$(1)$$

Within LPJ-LMfire, ignition efficiency (ieff) is a function dependent on fire danger calculated within the model (FDI), the area already burned to date in the grid cell ($ieff_{bf}$) and average ignition efficiency of the vegetation within each grid cell ($ieff_{avg}$). We, therefore, include the

effect of rising oxygen on ignition efficiency by scaling the ignition efficiency of each PFT using the equation above. Fuel moisture content within the fire module is divided into two categories: woody fuel moisture ($\omega_o$) and 1-h fuel and live grass moisture ($\omega_{nl}$), we therefore first calculate the ignition efficiency due to oxygen for each of the moisture contents:

$$ieff_{ox_g} = \frac{PI\left(O_2, \omega_{nl}\right)}{PI\left(20.95, \omega_{nl}\right)} \quad (2)$$

$$ieff_{ox_w} = \frac{PI\left(O_2, \omega_o\right)}{PI\left(20.95, \omega_o\right)} \quad (3)$$

Where $ieff_{ox_g}$ and $ieff_{ox_w}$ are the ignition efficiency due to oxygen for grasses and woody fuels respectively, normalised around present atmospheric levels of oxygen (20.95% vol. $O_2$) to produce a scaling factor. Ignition efficiency for each PFT ($ieff_{pft}$), based on prescribed constants (see Supplementary Table 2), is then scaled using the calculated values:

$$ieff_{pft_{ox}} = \begin{cases} ieff_{pft} \cdot ieff_{ox_g}, & pft = grass \\ ieff_{pft} \cdot ieff_{ox_w}, & pft = tree \end{cases} \quad (4)$$

Individual ignition efficiencies are then combined to give an average ignition efficiency due to oxygen and vegetation for each grid cell ($fpc_{grid}$), as a weighted average based on the vegetation foliar

projected cover (FPC) of each PFT:

$$\mathrm{ieff}_{avg} = \frac{\sum_{pft}^{npft}(\mathrm{fpc}_{grid} \cdot \mathrm{ieff}_{pft_{ox}})}{\sum_{pft}^{npft}\mathrm{fpc}_{grid}} \quad (5)$$

Overall ignition efficiency is then calculated using this weighted average:

$$\mathrm{ieff} = \mathrm{FDI} \cdot \mathrm{ieff}_{avg} \cdot \mathrm{ieff}_{bf} \quad (6)$$

## Moisture of extinction

Combustion experiments also show that as atmospheric oxygen concentration increases, the minimum fuel moisture content that prevents flame spread, termed moisture of extinction, also increases[30]. We, therefore, introduce an equation for moisture of extinction ($M_e$) as a function of atmospheric oxygen concentration ($O_x$) as outlined in Watson and Lovelock:[30]

$$M_e = 8O_x - 128 \quad (7)$$

We then calculate a moisture of extinction scaling factor ($M_{e\_ox}$), normalised around present-day oxygen levels (20.95% vol. $O_2$) that is used to scale existing moisture of extinction within the model to account for changing atmospheric oxygen concentrations:

$$M_{e\_ox} = \frac{M_e(O_2)}{M_e(20.95)} \quad (8)$$

## Heat of combustion

Heat of combustion, defined as the amount of energy released in a fire in the form of heat, has also been altered for our simulations. There is growing evidence to suggest that heat of combustion is dependent on variables such as fuel moisture, oxygen concentration, and also varies depending on the type of vegetation[33]. For instance, Babrauskas[33] states that much of the standard values used for heat of combustion are derived from oxygen-bomb-test values where fuels are burned under 100% oxygen which ensures complete combustion and that this is unrealistic and an overestimate as to the heat of combustion in a natural fire. In LPJ-LMfire, heat of combustion is set to a constant value of 18,000 KJg$^{-1}$ regardless of PFT distribution. However, data suggests that heat of combustion varies between the PFT groups, with the average being lower than this set value[33,35,36] (see Supplementary Table 1). We, therefore, improved heat of combustion (h) in the model, as an equation that is a function of atmospheric oxygen concentration ($O_x$) and varies depending on PFT group (see Supplementary Methods 1 for more detail):

$$h_{pft} = \frac{\alpha_{pft}}{O_x} + \beta_{pft} \quad (9)$$

Where $\alpha_{pft}$ and $\beta_{pft}$ are PFT coefficients and $h_{pft}$ is the heat of combustion for a given PFT (see Supplementary Table 2). Individual heat of combustion values are then combined to give a single weighted-average value for heat of combustion for each grid cell based on FPC of each PFT, replacing the single fixed value of 18000 KJg$^{-1}$:

$$h_{avg} = \frac{\sum_{pft}^{npft}\left(\mathrm{fpc}_{grid} \cdot h_{pft}\right)}{\sum_{pft}^{npft}\mathrm{fpc}_{grid}} \quad (10)$$

These changes to the fire module then affect other areas of fire behaviour such as rate of spread, which is shown to increase with rising oxygen concentrations[8,30].

## Simulation set up

Simulations were run with humans switched off (no anthropogenic ignitions, land-use etc.) so that only lightning-caused fires were present and input data sets used to drive the model are outlined in Supplementary Table 3. To analyse the effects on fire and vegetation, the model was run at varying levels of atmospheric oxygen concentration, ranging from present-day levels to 35% vol. $O_2$ in increments of 1%. We chose these bounds for our simulations in accordance with the range of validity of the combustion experiments conducted by Watson and Lovelock[30], from which we took equations for the probability of ignition and moisture of extinction. For all simulations the model was spun up for 1500 years, to ensure equilibrium is reached, with the last 10 years of the model analysed as output and plotted as 10-year annual averages using the R statistical software.

## Reporting summary

Further information on research design is available in the Nature Portfolio Reporting Summary linked to this article.

## Data availability

The raw LPJ-LMfire model data used in this study are available at https://github.com/ARVE-Research/LPJ_Oxygen-Fire. Whilst the Heat of Combustion data compiled for this study are provided in the Supplementary Information file.

## Code availability

The source code to run the version of LPJ-LMfire used for this research is archived at https://doi.org/10.5281/zenodo.7066216. Code and instructions for output analysis and creating the figures in this manuscript are available at https://github.com/ARVE-Research/LPJ_Oxygen-Fire.

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

## Acknowledgements

This research was funded by a Royal Society Research Professorship (A.J.W & R.V). We also acknowledge support from NERC NE/T003553/1 (C.M.B), previous support from a European Research Council Starter Grant that supported the conception of some of the ideas (ERC-2013-StG-335891- ECOFLAM)(C.M.B) and support by the BMBF-DAAD Make Our Planet Great Again Program (grant 57429870) (J.O.K).

## Author contributions

A.J.W., C.M.B and R.V. conceived the study. J.K. built the version of LPJLMfire upon which this study was based and provided the original code. R.V. found the required data and edited and ran the new model version. All authors contributed to the conception of the study and discussed the results. R.V. wrote the paper and all authors contributed to the writing and editing.

## Competing interests

The authors declare no competing interests.
