## [Peer Review File · Nature Communications]

Increased fire activity under high atmospheric oxygen concentrations is compatible with the presence of forestsREVIEWER COMMENTS

Reviewer #1 (Remarks to the Author):

Review of Vitali et al., "A new, higher limit for atmospheric oxygen compatible with large land plants"

Summary of key findings: It has been argued that atmospheric oxygen never rose above 30-35% at any time in Earth's history. This is because oxygen levels this high would lead to so many fires that forests could not be maintained (whereas the Phanerozoic geologic record documents continuous forests since the emergence of land plants). This study challenges this argument by varying atmospheric oxygen in a fire + dynamic vegetation model and calculating how steady state forest coverage changes as pO₂ increases. They find that while forest coverage is reduced at 36% pO₂ compared to 21% pO₂, the increased atmospheric oxygenation does not preclude the persistence of forests. This suggests that atmospheric oxygen levels could have varied dramatically over Phanerozoic time.

I will preface this review by saying my expertise is in Earth's long-term biogeochemical evolution and atmospheric oxygenation, not fire or vegetation modeling. I thus defer to other expert reviewers on the details of fire parameterizations and the implementation of dynamic global vegetation models. However, I can speak to the broader significance and persuasiveness of the paper. The general result—that oxygen levels above 30-35% are not incompatible with continuous forest coverage—is potentially of broad interest and suitable for publication in *Nature Communications*. However, the following issues should be addressed before publication:

While the modeling approach presented here is certainly novel and a major advance over previous studies, I'm not sure if it's quite accurate to state "previous calculations suggest an upper limit of around 30-35% O₂, based on assumptions that oxygen concentrations any higher would lead to widespread fires and prevent the regeneration of forests in the present world" (in the abstract, but similar statements are found elsewhere). For example, pages 120-123 of Berner's (2004) book point out the importance of fuel moisture in burning and do not seem to rule out oxygen above 30-35%. The GEOCARB error analysis in Royer et al. (2014; *AJS*) shows model O₂ values potentially exceeding 40% around 300 Ma (Fig. 2b). Statements about previous calculations should be adjusted accordingly.

Whether oxygen levels exceeding 30-35% could be attained depends on terrestrial primary productivity and organic carbon burial feedbacks. While the paper cites modeling work that explores this question, it would be informative if the paper addressed this more directly. For example, given that increasing O₂ from 21% to 36% results in a 26% reduction in forest cover, what would the implied change in terrestrial primary productivity and organic burial mean for the efficacy of fire-based feedbacks on atmospheric oxygen?

On a related note, Line 145 states that biogeochemical models such as COPSE (Bergman et al. 2004) assumed a 5% decrease in forest cover with a 20% → 35% increase in O₂. I'm not sure where this 5% number comes from. Looking at equations (7) and (8) in Bergman et al., increasing O' from 0.21 to 0.35 implies net primary productivity decreases by 45% (for $k_{fire} = 100$). A Lenton book chapter is also cited for the 5% number, but I do not have institutional access to this text. Lenton et al. (2018; *Earth-Science Reviews*) provides a different choice of parameters. In any case, a clearer explanation of fire feedbacks in the literature and how the results from this paper fit into that landscape would be beneficial. This could also potentially support the argument that oxygen levels could have exceeded 30-35%, despite the negative feedback from declining forest cover.

Perhaps what would be most useful here would be some example calculations showing that oxygen exceeding 30-35% is consistent with the demonstrated vegetation reduction (and other biogeochemical constraints/feedbacks). If the authors have access to COPSE (or GEOCARBSULF or similar), then some illustrative example outputs that incorporate a parameterization of their Fig. 2B and show that it is consistent with high oxygen would be compelling. I appreciate that this is

potentially asking a lot, and the authors should feel free to push back if this is a non-trivial undertaking better saved for a future publication. But if it could be implemented easily, then it would help make the paper even more impactful to the biogeochemical evolution community.

Why did the authors choose to stop their calculations at 36% oxygen? Are the experimental parameterizations not valid beyond this limit? Even if some extrapolation is necessary (and clearly delineated), it seems like running calculations at even higher pO₂ could be informative. For example, does the continuous persistence of forests not even rule out 40% or 50% oxygen? Is there perhaps no upper limit to the fire window? Some discussion of even higher O₂ levels would be informative.

The wording on lines 47-51 gives the impression that there is an ongoing debate over the 'fossil air' in amber claims, whereas my understanding was that these claims have been near-universally rejected by the scientific community, and were later recanted by Berner, one of the original proponents (Lane, 2002 "Oxygen"). If these historical claims are to be mentioned then they should be caveated more accurately.

Line 27 states "since the appearance of land plants approximately 420 million years ago, relatively little is known with certainty regarding the history of atmosphere oxygen (5)". To the extent that anything is known with certainty, this may be a slight overstatement. Phanerozoic oxygen evolution has received considerable attention in recent years (Royer et al. 2014, AJS; Mill et al. 2021, Gondwana Research; Lenton et al. 2018, Earth-Science Reviews etc.), and the oxygen lower limit since 420 Myr seems fairly robust, as explained in subsequent paragraphs. Phanerozoic oxygen is more certain than Proterozoic oxygen where proxy estimates span orders of magnitude!

Reviewer #2 (Remarks to the Author):

The manuscript 'A new, higher limit for atmospheric oxygen compatible with large land plants' reports the results of a new model for fire and forest growth/cover that considers fuel moisture content. Overall, the MS is well-written, the claims well supported, and I believe the subject and findings are of broad importance and thus suitable for publication in Nature Communications. Overall, this is a nice contribution that I'd like to see published in Nat. Comms. The manuscript is on the shorter side and I was left with a few questions that am hoping the authors can comment on in a modestly fleshed out version. Answers to some of these questions are likely speculative, but I think they should be addressed anyway. These include:

How does moisture content scale with climate? In other words, would your results hold in a warmer wetter world, or a colder drier world? Can you comment on this?

How does O₂ production scale with forest cover? Would expect an impact on O₂ production/ carbon (pyrite) burial at the diminished forest covers at say 35% O₂? Would this have an impact on the biosphere's ability to sustain a high O₂ atmosphere? I'd get that the answer to this question likely requires more modelling, but some comment would be useful.

What would set an upper limit on pO₂ if fire does not? Please comment.

Is there any data that can be used to support some of the assertions? For example, latitudinal gradients in charcoal distributions? Or any correlations between humidity and the charcoal record? ...or any other data of any kind that could tether the model results and claims to the real-world?

The MS wraps up rather abruptly. Could some additional information on implications and future directions be added?

Some minor comments below:

Ln 18-19 Something wrong with the last sentences of the abstract. Also, would be stronger with

specifics on what these implications are

Ln132-134 What is this based on? Please provide a reference or more context.

Ln136-138 Also here.

Ln 165 add a comma after behaviour

Ln 228 'show' should be 'shows', also add comma after 'simulation'

Ln 239 something wrong with grammar here

Ln 247 'period' should be 'periods'

Reviewer #3 (Remarks to the Author):

The title does not really reflect what is presented in the paper. The bulk of the text considers previous work on the interrelationships between vegetation, fire, moisture, ignition, oxygen concentration and suppression of forest regeneration. There are many factors to be considered and in this very short paper they are not satisfactorily justified.

For example, I could not extract from this manuscript why the oxygen concentration is considered by these authors to be the most important factor in suppressing (or not) forest regeneration. In the Results and Discussion section, the sentence (lines 166-170) confirms that oxygen concentration effects are complicated by fire frequencies and extent. Ignition is assumed to increase with oxygen concentration but there is no satisfactory reason or reasons provided for this. So many variables in fire behaviour (fuel, moisture, ignition etc.) need to be dealt with before the more tenuous assumption that oxygen concentration is so important, and so after running a series of experiments it can be concluded that the upper limit was likely to have been 35% vol.

There are simply too many "leaps of faith" in such a short paper that I still do not know whether or not to believe the results. This research needs a full length and fully discussed background to support the assumptions used, experimental design and result.

In other words, I do not think it is suitable for this journal.

RESPONSE TO REVIEWER COMMENTS

Reviewer #1 (Remarks to the Author):

Review of Vitali et al., "A new, higher limit for atmospheric oxygen compatible with large land plants"

We firstly thank the reviewer for taking time to review our manuscript and providing the following feedback which has greatly helped in improving the paper.

Summary of key findings: It has been argued that atmospheric oxygen never rose above 30-35% at any time in Earth's history. This is because oxygen levels this high would lead to so many fires that forests could not be maintained (whereas the Phanerozoic geologic record documents continuous forests since the emergence of land plants). This study challenges this argument by varying atmospheric oxygen in a fire + dynamic vegetation model and calculating how steady state forest coverage changes as pO₂ increases. They find that while forest coverage is reduced at 36% pO₂ compared to 21% pO₂, the increased atmospheric oxygenation does not preclude the persistence of forests. This suggests that atmospheric oxygen levels could have varied dramatically over Phanerozoic time.

I will preface this review by saying my expertise is in Earth's long-term biogeochemical evolution and atmospheric oxygenation, not fire or vegetation modeling. I thus defer to other expert reviewers on the details of fire parameterizations and the implementation of dynamic global vegetation models. However, I can speak to the broader significance and persuasiveness of the paper. The general result—that oxygen levels above 30-35% are not incompatible with continuous forest coverage—is potentially of broad interest and suitable for publication in Nature Communications. However, the following issues should be addressed before publication:

While the modeling approach presented here is certainly novel and a major advance over previous studies, I'm not sure if it's quite accurate to state "previous calculations suggest an upper limit of around 30-35% O₂, based on assumptions that oxygen concentrations any higher would lead to widespread fires and prevent the regeneration of forests in the present world" (in the abstract, but similar statements are found elsewhere). For example, pages 120-123 of Berner's (2004) book point out the importance of fuel moisture in burning and do not seem to rule out oxygen above 30-35%. The GEOCARB error analysis in Royer et al. (2014; AJS) shows model O₂ values potentially exceeding 40% around 300 Ma (Fig. 2b). Statements about previous calculations should be adjusted accordingly.

We thank the reviewer for highlighting this and statements throughout the text have been adjusted accordingly. For example, the abstract has been edited, discussion regarding the upper limit of oxygen has been expanded L83-99, the Royer paper brought to attention above has been included (L147) and comments regarding the importance of moisture content have been added (e.g. L270).

Whether oxygen levels exceeding 30-35% could be attained depends on terrestrial primary productivity and organic carbon burial feedbacks. While the paper cites modeling work that explores this question, it would be informative if the paper addressed this more directly. For

example, given that increasing O₂ from 21% to 36% results in a 26% reduction in forest cover, what would the implied change in terrestrial primary productivity and organic burial mean for the efficacy of fire-based feedbacks on atmospheric oxygen?

In light of this and another reviewer's comment, both the introduction and discussion have been expanded to include literature and thoughts on fire feedbacks. A paragraph introducing the importance of fire feedbacks on atmospheric oxygen has been added from L101-L128. Whilst discussion surrounding the comments above directly have been added from L341 in the following:

"Whilst the upper limit of the fire window is higher than previously thought, the upper limit of atmospheric oxygen itself and whether forests can exist under high oxygen concentrations over the Earth's history, relies on more complex mechanisms and the feedbacks at play. For instance, decreases in forest biomass under high atmospheric oxygen concentrations directly limits the amount of organic carbon available for carbon burial, the long-term source of O₂ (5, 32, 39). Several negative fire feedbacks have been proposed that also lower rates of carbon burial under high oxygen levels such as reduced phosphorus weathering by roots as a result of fire suppression on vegetation, which in turn lowers productivity and hence carbon burial (5). It is therefore possible that the 60% reduction in forest cover under 35% vol. O₂ (compared to no fire forest cover) is enough to slow carbon burial to an extent that the rise of atmospheric oxygen is too slow compared to the measures that counteract it, limiting the ability of the biosphere to sustain a high oxygen atmosphere. Whether or not atmospheric oxygen can surpass levels of 30/35% vol. O₂ thus relies on the strength of such geochemical feedbacks on oxygen concentrations. Other non-fire linked processes influence the concentration of atmospheric oxygen through time and may therefore have a more important role than previously considered. High atmospheric oxygen can limit productivity of plants through inhibiting CO₂ fixation through the Rubisco enzyme (73, 74) which would further impact vegetation biomass and hence carbon burial. Other processes have the potential to counteract reductions in organic carbon burial under high atmospheric levels, such as continental uplift which can enhance carbon burial through increasing the flux of reactive phosphorus (32). Periods of increased uplift could therefore lead to high atmospheric oxygen levels greater than 30% vol. O₂ being reached, as may have been the case during the Permian-Carboniferous (75)..."

On a related note, Line 145 states that biogeochemical models such as COPSE (Bergman et al. 2004) assumed a 5% decrease in forest cover with a 20% → 35% increase in O₂. I'm not sure where this 5% number comes from. Looking at equations (7) and (8) in Bergman et al., increasing O' from 0.21 to 0.35 implies net primary productivity decreases by 45% (for k_{fire} = 100). A Lenton book chapter is also cited for the 5% number, but I do not have institutional access to this text. Lenton et al. (2018; Earth-Science Reviews) provides a different choice of parameters.

We apologise here for any misunderstanding. Whilst the reviewer is correct in that much larger decreases in biomass/cover is seen from PAL (~21% O₂) to high atmospheric oxygen of around 35% vol. O₂, the value of 5% came from what is assumed to be the current suppression of biomass from fire under PAL compared to a no-fire scenario (i.e. <17% vol. O₂). This value essentially determines what the 'k_{fire}' value is within the models which is a parameter that is a set value which reflects how strong fire feedbacks are assumed to be. For instance, the k_{fire} = 100 case mentioned above is regarded as a 'weak fire-feedback' in which

with no-fire vegetation at PAL increases by 1%. Whilst frequently models now adapt $k_{\text{fire}} = 3$ in simulations in accordance to Bond et al. (2005) results which suggests vegetation under PAL is 50% suppressed by fire (strong feedback) (a nice explanation of this can be found in the methods of Belcher et al. (2021) 'rise of angiosperms' paper).

We agree however that this wasn't entirely clear and so the text has been edited to try and improve clarity on this through the following:

This result implies that not only could the upper limit of the fire window be higher than previously considered, but also fire feedbacks to atmospheric oxygen may be weaker. We found that the fire suppression on present-day forest cover is approximately 26% when compared to a world without fire case (<17% vol. O₂), almost half of the 50% reduction estimated by Bond et al. (51) who found that present-day forest cover would double in a world without fire. Yet still significantly greater than initial estimates of 5% reduction of vegetation biomass under PAL compared to the 'no-fire' scenario used in previous versions of biogeochemical models (5, 36), supporting that the effects of fire feedbacks on atmospheric oxygen concentration may be substantial.

In any case, a clearer explanation of fire feedbacks in the literature and how the results from this paper fit into that landscape would be beneficial. This could also potentially support the argument that oxygen levels could have exceeded 30-35%, despite the negative feedback from declining forest cover.

As above, the manuscript now has explicit discussion of fire feedbacks from literature and relevance to our results both in the introduction (L101) and in the discussion.

Perhaps what would be most useful here would be some example calculations showing that oxygen exceeding 30-35% is consistent with the demonstrated vegetation reduction (and other biogeochemical constraints/feedbacks). If the authors have access to COPSE (or GEOCARBSULF or similar), then some illustrative example outputs that incorporate a parameterization of their Fig. 2B and show that it is consistent with high oxygen would be compelling. I appreciate that this is potentially asking a lot, and the authors should feel free to push back if this is a non-trivial undertaking better saved for a future publication. But if it could be implemented easily, then it would help make the paper even more impactful to the biogeochemical evolution community.

We thank the reviewer for their interesting point and is something we hope to do in the future.

Why did the authors choose to stop their calculations at 36% oxygen? Are the experimental parameterizations not valid beyond this limit? Even if some extrapolation is necessary (and clearly delineated), it seems like running calculations at even higher pO₂ could be informative. For example, does the continuous persistence of forests not even rule out 40% or 50% oxygen? Is there perhaps no upper limit to the fire window? Some discussion of even higher O₂ levels would be informative.

The range of atmospheric oxygen levels simulated were chosen due to the range of validity of the equations taken from the combustion experiments by Watson and Lovelock (2013) that we incorporated into the model. Hence running the model at greater levels as is would be invalid and extrapolation of the equation would pose large uncertainties and have limited value. From this we have added the following sentence into the methods:

“Simulations were run between these bounds in accordance with the range of validity of the combustion experiments conducted by Watson and Lovelock (30), from which we took equations for the probability of ignition and moisture of extinction.”

We have however, added some discussion regarding higher levels of oxygen and the upper limits of the fire window. For instance, L334:

“Furthermore, the suppression of forest cover by fire appears to be stabilising under high oxygen concentrations (Fig. 2B) and so at levels greater than 35% vol. O₂ we would expect the impact of fire to have a limited further effect on forest cover as fuel moisture would still be the limiting factor. We argue therefore, that the upper limit of the fire window is likely much higher than 35% vol. O₂ or possibly unbounded.”

The wording on lines 47-51 gives the impression that there is an ongoing debate over the ‘fossil air’ in amber claims, whereas my understanding was that these claims have been near-universally rejected by the scientific community, and were later recanted by Berner, one of the original proponents (Lane, 2002 “Oxygen”). If these historical claims are to be mentioned then they should be caveated more accurately.

We appreciate the reviewers point, this section has now been revised to clarify the scrutiny and rejection of the measurements through the following:

“The closest we have come to direct evidence of high oxygen concentration is in the form of ‘fossil air’ trapped in amber samples from the Cretaceous period, which measured 30% vol. O₂ (14). However, the samples have since been widely scrutinised (15), for instance concerns as to how effectively amber bubbles can create an air-tight seal, potentially causing distortion in the composition of the samples over long time periods, leading to a near-universal rejection of such measurement (15, 16)”

Line 27 states “since the appearance of land plants approximately 420 million years ago, relatively little is known with certainty regarding the history of atmosphere oxygen (5)”. To the extent that anything is known with certainty, this may be a slight overstatement. Phanerozoic oxygen evolution has received considerable attention in recent years (Royer et al. 2014, AJS; Mill et al. 2021, Gondwana Research; Lenton et al. 2018, Earth-Science Reviews etc.), and the oxygen lower limit since 420 Myr seems fairly robust, as explained in subsequent paragraphs. Phanerozoic oxygen is more certain than Proterozoic oxygen where proxy estimates span orders of magnitude!

We thank the reviewer for highlighting this point and have adjusted the line to the following:

"However, whilst estimates on the lower limit of atmospheric oxygen since the appearance of land plants approximately 420 million years ago (mya) are fairly robust (8–13), estimates of the upper limit of atmospheric oxygen and processes involved are still poorly understood (5). "

Reviewer #2 (Remarks to the Author):

We firstly wish to thank the reviewer for their time and feedback on our manuscript which has enabled further improvement.

The manuscript 'A new, higher limit for atmospheric oxygen compatible with large land plants' reports the results of a new model for fire and forest growth/cover that considers fuel moisture content. Overall, the MS is well-written, the claims well supported, and I believe the subject and findings are of broad importance and thus suitable for publication in Nature Communications. Overall, this is a nice contribution that I'd like to see published in Nat. Comms. The manuscript is on the shorter side and I was left with a few questions that am hoping the authors can comment on in a modestly fleshed out version. Answers to some of these questions are likely speculative, but I think they should be addressed anyway. These include:

How does moisture content scale with climate? In other words, would your results hold in a warmer wetter world, or a colder drier world? Can you comment on this?

A brief paragraph discussing this together with geological evidence has been added from 292. For example:

"From our results, we would expect that in a warmer, wetter world forests would be less flammable globally than a colder drier world as the flammability of forests would be limited through fuel moisture content. This is demonstrated by the observed reduction in Eocene charcoal compared to the preceding Palaeocene (17, 18) has been attributed to a hydrological change and increased rainfall (68) which led to widespread rainforest biomes that suppressed fire (18) ... "

How does O₂ production scale with forest cover? Would expect an impact on O₂ production/ carbon (pyrite) burial at the diminished forest covers at say 35% O₂? Would this have an impact on the biosphere's ability to sustain a high O₂ atmosphere? I'd get that the answer to this question likely requires more modelling, but some comment would be useful.

Considering this and another reviewer's comment, both the introduction and discussion have been expanded to include literature and thoughts on fire feedbacks on atmospheric oxygen and vegetation against results found. Specifically, discussion has been added from L341 in the following:

"Whilst the upper limit of the fire window is higher than previously thought, the upper limit of atmospheric oxygen itself and whether forests can exist under high oxygen

concentrations over the Earth's history, relies on more complex mechanisms and the feedbacks at play. For instance, decreases in forest biomass under high atmospheric oxygen concentrations directly limits the amount of organic carbon available for carbon burial, the long-term source of O₂ (5, 32, 39). Several negative fire feedbacks have been proposed that also lower rates of carbon burial under high oxygen levels such as reduced phosphorus weathering by roots as a result of fire suppression on vegetation which in turn lowers productivity and hence carbon burial (5). It is therefore possible that the 60% reduction in forest cover under 35% vol. O₂ (compared to no fire forest cover) is enough to slow carbon burial to an extent that the rise of atmospheric oxygen is too slow compared to the measures that counteract, limiting the ability of the biosphere to sustain a high oxygen atmosphere. Whether or not atmospheric oxygen can surpass levels of 30/35% vol. O₂ thus relies on the strength of such fire feedbacks on oxygen. Other non-fire linked processes influence the concentration of atmospheric oxygen through time and may therefore have a more important role than previously considered. High atmospheric oxygen can limit productivity of plants through inhibiting CO₂ fixation through the Rubisco enzyme (74, 75) which would further impact vegetation biomass and hence carbon burial. Other processes have the potential to counteract reductions in organic carbon burial under high atmospheric levels, such as continental uplift which can enhance carbon burial through increasing the flux of reactive phosphorus (32). Periods of increased uplift could therefore lead to high atmospheric oxygen levels greater than 30% vol. O₂ being reached, as may have been the case during the Permian-Carboniferous (76)..."

What would set an upper limit on pO₂ if fire does not? Please comment.

Whilst the research presented suggests that fires effect on vegetation and hence atmospheric oxygen is weaker than previously assumed, it is likely still strong enough to set an upper limit through fire feedbacks. Under high levels of O₂ it seems moisture plays a key role, but other mechanisms or processes that could also help to regulate atmospheric oxygen are still largely unknown. Comments on this have been added to the discussion. For example, the added paragraph starting from L341 touches on this, whilst the concluding paragraph has been edited and includes:

"Our research shows that the assumptions on the upper limit of the fire window do not hold in that forests are able to exist in the present world under high atmospheric oxygen levels (>30% vol. O₂) and fuel moisture is likely important under high concentrations, limiting suppression of vegetation by fire. This suggests we have a limited understanding of the controls of atmospheric oxygen through time. Whilst it seems that fire plays a lesser role in setting an upper limit of atmospheric oxygen, other processes of importance are still largely unknown and so new mechanisms must be sought"

Is there any data that can be used to support some of the assertions? For example, latitudinal gradients in charcoal distributions? Or any correlations between humidity and the charcoal record? ...or any other data of any kind that could tether the model results and claims to the real-world?

We appreciate here that data that can directly support the assertions made here would be ideal. However, we have found there to be minimal data that would be relevant to the study

such as latitudinal gradients or studies with data relating humidity to the charcoal record. Though considering this, the paragraph from L292 attempts to place assertions made against observations made in the real world.

The MS wraps up rather abruptly. Could some additional information on implications and future directions be added?

We thank the reviewer for bringing this to our attention. The last paragraphs (from L391) of the manuscript has been significantly expanded to include more implications and future directions.

Some minor comments below:

Ln 18-19 Something wrong with the last sentences of the abstract. Also, would be stronger with specifics on what these implications are

Corrections made to the last sentence and more specifics added on last line of abstract to read:

“This implies that the effect of fires on suppressing global vegetation is lower than previously assumed which questions our understanding of the mechanisms involved in regulating the abundance of oxygen in our atmosphere and highlights moisture as a potentially important factor.”

Ln132-134 what is this based on? Please provide a reference or more context.

More background and justification has been added regarding moisture of extinction including more references. (L189-199):

“Whether an ignition can lead to sustained combustion (e.g. spread) is determined by various factors including temperature, fuel moisture, wind speed, fuel density etc. (56, 59). Of these, moisture content is thought to play a critical role (10, 59). Whilst dry fuels enable fire spread, if fuel moisture content surpasses a certain threshold, fire cannot be sustained - defined as the moisture of extinction (60, 61) and is included in calculations within the fire module of LPJ-LMfire (56, 58). The value of moisture of extinction for different natural fuels has been found to increase with oxygen concentration in combustion experiments (10, 30). We therefore extend calculations of moisture of extinction within the LPJ-LMfire model to include changes over atmospheric oxygen concentration following combustion experiments conducted by Watson and Lovelock (30) (See Methods).”

Ln136-138 Also here.

Similarly more context is given regarding the heat of combustion to include more background and references from L201-208:

“Finally, the concentration of atmospheric oxygen can also influence the energy released from a fire (62, 63). Heat of combustion (HoC) is defined to be the total amount of energy released in a fire in the form of heat (64) and has been found to vary across different vegetation types (62, 64, 65). We therefore alter HoC within LPJ-LMfire so that each plant functional type has its own discrete HoC value as opposed to a set constant across all vegetation. Furthermore, studies suggest that HoC also varies across different oxygen concentrations (62, 63) and so we also derive a function that enables atmospheric oxygen concentration to influence the value of HoC for each of the vegetation types within the model (see Methods and SM1).”

Ln 165 add a comma after behaviour – **Corrected.**

Ln 228 ‘show’ should be ‘shows’, also add comma after ‘simulation’ – **Corrected.**

Ln 239 something wrong with grammar here – **Our apologies here, grammar has been corrected.**

Ln 247 ‘period’ should be ‘periods’ – **Corrected**

Reviewer #3 (Remarks to the Author):

We thank the reviewer for their time reading and reviewing the manuscript. Although we are sorry to hear that the reviewer does not consider it suitable, we are grateful for the comments which have led to an overall improvement of the manuscript.

The title does not really reflect what is presented in the paper – **the title has been changed to more accurately reflect the research presented**

The bulk of the text considers previous work on the interrelationships between vegetation, fire, moisture, ignition, oxygen concentration and suppression of forest regeneration. There are many factors to be considered and in this very short paper they are not satisfactorily justified.

We thank the reviewer for bringing this to the author’s attention. In response, both the introduction and discussion have been significantly expanded. For example: L34-45 provides more background into the history of oxygen and fire, L74-83 provides further detail on previous experimental studies, L101-128 includes background on feedbacks between fire and oxygen, whilst 166-208 have been significantly added too to provide justification for our methods.

For example, I could not extract from this manuscript why the oxygen concentration is considered by these authors to be the most important factor in suppressing (or not) forest regeneration.

Our reason for considering atmospheric oxygen to be of large importance stems from the fire triangle and the fact it compromises one of the three key elements for a fire to occur and that the manuscript is centred around the upper limit of the fire window: the range of atmospheric oxygen concentrations valid for fire. Previous assumptions on the upper limit of the fire windows have relied on the assumption that at levels above ~35% vol. O₂ fires become so widespread that fires threaten forest regeneration. We are testing this assumption.

To try to justify the importance, the introduction has been expanded. For example L34-45 now describe background between fire and oxygen:

“The presence of fossil charcoal in sedimentary rocks since the late Silurian has been used not only to indicate the occurrence of wildfires throughout subsequent evolutionary history, but also to put constraints on the variability of atmospheric oxygen (12, 13). For an ignition and sustained fire to occur, three basic elements are required: an ignition or heat source, fuel that can burn and a supply of oxygen (14). It is likely that natural ignition sources have always been present on Earth (15), with lightning strikes being continuously present on our planet since the Ordovician (5) and adequate fuel has been available since the evolution of land plants around 420 mya (12). Thus any fire occurring from then indicates that atmospheric oxygen concentration must have fallen within the bounds that support natural fire, termed the “fire window” (6, 16). Subsequently, fluctuations of atmospheric oxygen levels are accompanied by fluctuations of the flammability of our planet throughout time (6, 17, 18). The lower limit of the fire window has been investigated in numerous experimental studies (5–10)”

In the Results and Discussion section, the sentence (lines 166-170) confirms that oxygen concentration effects are complicated by fire frequencies and extent. Ignition is assumed to increase with oxygen concentration but there is no satisfactory reason or reasons provided for this.

The assumption that the probability of ignition increases sharply with atmospheric oxygen concentration comes from several studies which have found so through combustion experiments on natural fuels which are references in the introduction as well as the methods. The introduction discussion on the combustion experiments have been more explicitly discussed to clarify this point.

So many variables in fire behaviour (fuel, moisture, ignition etc.) need to be dealt with before the more tenuous assumption that oxygen concentration is so important, and so after running a series of experiments it can be concluded that the upper limit was likely to have been 35% vol.

For this study we opted to use this version of the LPJ DGVM since it has a robust process-based fire module coupled to it, meaning all the aspects of fire behaviour mentioned above

are simulated reasonably within the model. Whilst we agree such variables are incredibly important for fire behaviour, the aim of this research was to test the long-standing assumptions of the fire window: that rising oxygen concentrations result in decreasing and eventually no forest cover, and so are looking at the importance of the relationship between fire and oxygen concentration. Furthermore, there are numerous pieces of research that support that oxygen concentration is important regarding the probability of ignition or rate of fire spread (for instance combustion experiments discussed in manuscript). We have also specifically looked at how oxygen concentration alters the fire behaviour variables you mention such as moisture of extinction and probability of ignition. We therefore feel that we have covered this, however we have added a lengthier discussion to try and clarify as above.

There are simply too many "leaps of faith" in such a short paper that I still do not know whether or not to believe the results. This research needs a full length and fully discussed background to support the assumptions used, experimental design and result. In other words, I do not think it is suitable for this journal.

REVIEWERS' COMMENTS

Reviewer #1 (Remarks to the Author):

The authors have thoroughly addressed my comments and questions. The revised manuscript is suitable for publication in its current form.

Reviewer #3 (Remarks to the Author):

The authors have mostly addressed my concerns and put a lot of effort in responding to all the reviewers.

Response to Reviewers

REVIEWERS' COMMENTS

Reviewer #1 (Remarks to the Author):

The authors have thoroughly addressed my comments and questions. The revised manuscript is suitable for publication in its current form.

Reviewer #3 (Remarks to the Author):

The authors have mostly addressed my concerns and put a lot of effort in responding to all the reviewers.

We thank the reviewers for their time and are grateful for their comments.